# Is There a Causal Relationship between Childhood Obesity and Acute Lymphoblastic Leukemia? A Review

**DOI:** 10.3390/cancers12113082

**Published:** 2020-10-22

**Authors:** Molly J. Dushnicky, Samina Nazarali, Adhora Mir, Carol Portwine, Muder Constantine Samaan

**Affiliations:** 1Department of Pediatrics, McMaster University, Hamilton, ON L8N 3Z5, Canada; molly.dushnicky@medportal.ca (M.J.D.); samina.nazarali@medportal.ca (S.N.); adhora.mir@medportal.ca (A.M.); portwc@mcmaster.ca (C.P.); 2Division of Pediatric Endocrinology, McMaster Children’s Hospital, Hamilton, ON L8N 3Z5, Canada; 3Michael G. De Groote School of Medicine, McMaster University, Hamilton, ON L8S4L8, Canada; 4Division of Pediatric Hematology/Oncology, McMaster Children’s Hospital, Hamilton, ON L8N 3Z5, Canada; 5Department of Health Research Methods, Evidence and Impact, McMaster University, Hamilton, ON L8S 4K1, Canada

**Keywords:** childhood obesity, acute lymphoblastic leukemia, adipocyte, natural killer cells, cytokines, adipose stem cells

## Abstract

**Simple Summary:**

The childhood obesity epidemic is impacting tens of millions of children globally. While obesity causes several cancers in adults, its potential role in causing pediatric cancers remains unclear. In this review, we assess the potential contribution of obesity to the development of acute lymphoblastic leukemia (ALL), the most common pediatric cancer. We review the possible mechanisms by which the adipose tissue attracts and protects leukemia cells and how it interferes with the actions of chemotherapies used in ALL treatment. We also examine adipose tissue-secreted molecules and fuels that may support leukemia development. While there are no current definite causal links between obesity and ALL, there are plausible mechanisms that need further investigation to explore the impact of obesity on causing ALL and on impacting treatment outcomes.

**Abstract:**

Childhood obesity is a growing epidemic with numerous global health implications. Over the past few years, novel insights have emerged about the contribution of adult obesity to cancer risk, but the evidence base is far more limited in children. While pediatric patients with acute lymphoblastic leukemia (ALL) are at risk of obesity, it is unclear if there are potential causal mechanisms by which obesity leads to ALL development. This review explores the endocrine, metabolic and immune dysregulation triggered by obesity and its potential role in pediatric ALL’s genesis. We describe possible mechanisms, including adipose tissue attraction and protection of lymphoblasts, and their impact on ALL chemotherapies’ pharmacokinetics. We also explore the potential contribution of cytokines, growth factors, natural killer cells and adipose stem cells to ALL initiation and propagation. While there are no current definite causal links between obesity and ALL, critical questions persist as to whether the adipose tissue microenvironment and endocrine actions can play a causal role in childhood ALL, and there is a need for more research to address these questions.

## 1. Introduction

The rates of childhood obesity have started to plateau in high-income countries, yet they continue to rise in low- and middle-income countries [1]. On a global scale, the obesity epidemic impacts one in three people, including tens of millions of children. These staggering figures are coupled with evidence that obesity management programs have modest success rates, and prevention remains the best cure [2,3,4,5]. Managing and preventing obesity may help lower its cardiometabolic and psychological comorbidities that, at times, start during childhood and persist into adulthood [6,7,8,9,10,11,12,13,14].

In adults, obesity has been linked to the increased risk of several malignancies, including esophageal adenocarcinoma, colon, renal cell, uterine and postmenopausal breast cancers [15,16,17,18]. On the other hand, a normal body mass index (BMI, <25 kg/m^2^) is associated with reduced rates of gastric cardia, gallbladder, pancreatic, ovarian, and thyroid cancer, as well as meningiomas, hepatocellular carcinoma and multiple myeloma [15,19,20,21,22,23,24,25]. Notably, obesity during childhood is associated with an increased risk of adult pancreatic and colon cancers, especially in women [26,27,28]. The evidence for the association of childhood obesity with other adult and pediatric cancers is scarce.

The most common pediatric cancer is acute lymphoblastic leukemia (ALL) accounting for almost 25% of all childhood cancers. ALL is also responsible for 26% of cancer-related mortality, making it the second most common driver of cancer-related deaths after brain tumors [15,26,29].

While there is no current definite causal link between obesity and ALL, there are several plausible mechanisms and parallels whereby obesity may contribute to ALL development and outcomes.

Several models have demonstrated that the adipose tissue supports the rapid progression of certain cancers. For example, leptin and resistin, two adipose tissue products, promoted the growth of melanoma cells and reduced the therapeutic efficacy of dacarbazine, the chemotherapeutic agent used to treat it. Diet-induced obesity promoted melanoma development and reduced dacarbazine ability to access melanoma cells [30,31]. These lines of evidence point to the potential contribution of the adipose tissue and its products to carcinogenesis.

In this review, we evaluate the evidence for the association of obesity with ALL at diagnosis and end of therapy, and its links to the risk of relapse and mortality. We also review potential Immunometabolic and endocrine mechanisms that may promote ALL development and propagation in obese children. Finally, we propose future research priorities to further the understanding of the impact of obesity on ALL genesis and outcomes.

## 2. The Association of Childhood Obesity & Birth Weight with ALL

### 2.1. Obesity at Initial ALL Diagnosis

While the rates of pediatric obesity and ALL have risen over the past few decades [32,33], most patients with ALL have normal BMI values at diagnosis. However, weight increases are common during and after ALL treatment (Table 1) [34,35,36,37,38,39,40,41].

In addition, a few patient subgroups may be at risk of obesity when compared to the whole ALL cohort. Early evidence suggested that obesity rates are increased among children aged 5–9 years and females at diagnosis of ALL when compared to other age groups and males, respectively [34,35,42]. More recent studies demonstrated higher obesity rates in males, children with B-cell ALL, and in those with central nervous system involvement [38]. Importantly, survivors of childhood ALL who were obese at diagnosis were more likely to be obese as adults [37]. In summary, body mass increases from diagnosis to the end of therapy and beyond are common in survivors of childhood ALL. A deeper understanding of why prepubertal children, ALL subtype, and central involvement are associated with increased obesity risk is needed. The impact of sex on ALL is unclear and requires further study. Regardless, obesity during childhood is likely to propagate into adult life which may drive type 2 diabetes and cardiovascular disease in survivors.

### 2.2. Birth Weight and ALL

An elevated birth weight has been linked to an increased risk of ALL in children (Table 2).

Chromosomal translocations such as t(12:21), t(4:11) or t(8:21) contribute to the uncontrolled proliferation of lymphoblasts, and high birth weight may drive this proliferation with high levels of growth factors [43,44,46,48,49,50,51]. Furthermore, having a high birth weight when corrected for gestational age and rapid fetal growth rates predict ALL risk during childhood [45,47]. In summary, birth weight, a variable determined by multiple prenatal influences, may play a role in the development of ALL, and the exact mechanisms by which birth weight affects ALL risk require further interrogation.

## 3. Obesity in Childhood ALL Survivors and Its Effects on Outcomes

### 3.1. Obesity is Common Among Survivors of Childhood ALL

Treatment of childhood ALL increases the risk of obesity in survivors when compared to levels noted at diagnosis (Table 1). The weight gain starts during treatment, and 40–50% of young adult survivors remain obese at follow-up [36,42]. The most significant weight gain occurs between maintenance phases 1–3, approximately 9–12 months into treatment protocols [35,39,52].

While some studies suggested that age, sex, cranial irradiation (CRT) and steroids may play a role in obesity development in ALL survivors [35,37,40,41,53,54,55], a recent meta-analysis of 20 studies (*n* = 1742) demonstrated that the mean BMI z-score was 0.83 (95% CI 0.60–1.06). This z-score corresponds to the 80th percentile, which is significantly higher than reference populations, including the National Health and Nutrition Examination Survey (NHANES; BMI z-score 0.4–0.6). While studies had high heterogeneity, subgroup analyses revealed a high prevalence of obesity in ALL survivors (29–69% 5–9 years post-treatment; 34–46% >10 years post-treatment) but obesity risk was not consistently linked to age, female sex, CRT, and steroid type or dose [56].

The obesogenic impact of chemotherapeutic agents has not been widely assessed. However, high-intensity induction chemotherapy (the addition of anthracycline ± cyclophosphamide) does not seem to affect BMI z-scores above standard induction chemotherapy (steroids, vincristine and l-asparaginase) [55]. In summary, there are likely factors beyond age, sex, and treatments that drive obesity risk in ALL survivors, and these factors require identification and further analysis.

### 3.2. The Association of Obesity with Relapse Risk and Mortality in ALL

Obesity may impact treatment outcomes in ALL. Overweight and obese children have worse event-free survival and a higher rate of morbidities than non-obese children during the induction phase [57,58,59,60]. Furthermore, obesity during the premaintenance chemotherapy phase is a risk factor for hypertension (OR 3.27; 95% CI 1.10–10.00; *p* = 0.05), hyperglycemia (OR 2.62; 95% CI 1.04–6.56; *p* = 0.04), and febrile neutropenia (incident rate ratio 1.53; 95% CI 1.10–2.12; *p* = 0.01) [60]. Obesity is also a risk factor for persistent minimal residual disease following induction chemotherapy (OR 2.57; 95% CI 1.19–5.54; *p* = 0.016) [59].

Early evidence suggested that an elevated body mass was a risk factor for ALL relapse [61,62]. However, recent meta-analyses demonstrated a nonsignificant trend in overweight and obese children [58,63] (Table 3). These data suggest that obesity does not drive relapse risk in ALL.

On a mechanistic level, in-vitro and in-vivo evidence suggests that adipocytes may affect leukemia treatment efficacy by attracting and supporting the proliferation of lymphoblasts, and there are several protective types of machinery against leukemia cell destruction [64].

## 4. Potential Mechanisms by Which Obesity May Contribute to ALL Pathogenesis

The adipose tissue is regarded as one of the most metabolically active organs in the body [65,66,67,68], and some of its endocrine actions are mediated via compounds called adipocytokines. Adipose tissue expansion in obesity is associated with immune system activation and inflammation that promotes several cardiometabolic disorders such as atherosclerosis, hyperglycemia and dyslipidemia [69,70,71].

While there are data that link obesity as a causal factor to certain adult cancers [15], there is no direct evidence to suggest that obesity causes ALL in children. However, a recent study demonstrated that high-fat-fed mice experience a more rapid progression of ALL than lean mice, offering a potential link between obesity and ALL evolution [72]. Obesity is also associated with the production of several adipocytokines hypothesized to promote oncogenesis, including leptin, tumor necrosis factor-alpha (TNFα), interleukin-6 (IL-6), IL-7, and IL-8 [73,74,75,76,77]. Below, we discuss some putative pathways in which the adipose tissue may contribute to ALL pathogenesis (Figure 1).

The adipocytokine secretion by the obese adipose tissue is altered, with increased secretion of leptin and reduced production of adiponectin. The changes in these adipokines alter leukemia cell activity, cytokine secretion and bone marrow angiogenesis. Increased secretion of cytokines, including IL-7 and IL-3, promotes the survival and proliferation of leukemia cells through the activation of the phosphatidylinositol 3-kinase (PI3K) and mitogen-activated protein kinase (MAPK) pathways. The upregulated production of growth factors such as insulin and insulin-like growth factor-1 (IGF-1) contributes to enhanced chemotherapy resistance and drives poor prognosis through the overexpression of Janus kinase/signal transducer and activator of transcription protein 3 (JAK/STAT3) and activation of the PI3K and MAPK pathways. Obesity also contributes to increased adipocyte-derived fuel production of glucose, free fatty acids and amino acids, which causes leukemia cell proliferation (Figure developed in BioRender, https://biorender.com/).

### 4.1. White Adipose Tissue is Far More Than a Passive Calorie Sink in Obesity

The adipose tissue is comprised of three main subtypes, including white adipose tissue (WAT), beige adipose tissue and brown adipose tissue [78]. WAT can be further subdivided based on depot location into visceral or subcutaneous compartments. The former depot secretes inflammatory cytokines in obesity while the latter depot stores excess triglycerides and releases free fatty acids (FFAs) throughout periods of starvation, fasting or exercise. While brown adipose tissue is most prominent in infants and decreases with age, WAT increases with aging and obesity. The expansion of visceral WAT is associated with the risk of cardiovascular diseases, metabolic syndrome, and cancer [79,80,81,82,83].

In obesity, adipose tissue expansion leads to relative tissue hypoxia as the tissue demands exceed vascular supply and oxygenation [84]. In turn, hypoxia upregulates angiogenesis and the production of chemokines that attract immune cells into the adipose tissue [85]. The main source of cytokines in the obese adipose tissue are infiltrating immune cells, including macrophages, and these cytokines drive local tissue inflammation and insulin resistance. The ability of the adipose tissue to expand and act as a reservoir for cytokines and FFAs is finite. These molecules exit the adipose tissue, enter the circulation, and arrive at skeletal muscle and liver where they drive inflammation and insulin resistance in these metabolic organs [86,87,88,89,90].

### 4.2. Adipocytes Interference in Pharmacokinetics of ALL Chemotherapies

There is evidence that excess adipose tissue may lower ALL treatment efficacy by attracting lymphoblasts, supporting their proliferation and protecting them from destruction through several mechanisms (Figure 1) [64]. Lymphoblasts have been identified within the adipose tissue of mice with ALL, suggesting that these cells migrate into the adipose compartment [91]. The retro-orbital transplantation of syngeneic leukemia cells into lean and high-fat-fed obese mice resulted in the migration of these cells to the adipose tissue in-vivo [64]. Cells were also detected in the bone marrow, spleen, liver and visceral and perirenal fat. Obese mice had more leukemia cells per milligram of visceral fat than nonobese mice. The adipocyte stromal cell-derived factor 1-α (SDF-1α), also known as cysteine-X-cysteine (C-X-C) motif chemokine 12 (CXCL12), promoted leukemia cell migration into the adipose tissue [64].

Adipose tissue also alters the pharmacokinetics of chemotherapeutic agents via a combination of mechanisms including the increased accumulation of lipid-soluble drugs, increased binding of basic chemotherapeutic agents due to elevated levels of α1-acid glycoproteins and increased secretion of water-soluble drugs through activation of cytochrome P450 2E1 (Figure 1) [92]. For example, leukemia cells have an impaired response to vincristine, a lipophilic anti-leukemia agent, in obese mice. While blood and tissue vincristine levels were similar in obese and lean mice, the leukemia cells, when cocultured with adipocytes in vitro, migrated beneath the adipocyte layer, which led to reduced vincristine exposure [91].

A lower serum concentration of 6-mercaptopurine (6-MP), a drug used to maintain remission in ALL patients, was noted with a BMI >75th percentile when compared to children with a BMI <75th percentile [93]. Adipocytes also sequester and metabolize other chemotherapeutic agents, including anthracyclines, daunorubicin and doxorubicin, through the deactivating enzymes aldo-keto reductases and carbonyl reductases [94]. The adipocytes also upregulate pro-survival signals, including B-cell lymphoma 2 (BCL-2) and serine/threonine-protein kinase Pim-2 (PIM2). Furthermore, adipocytes inactivate pro-apoptotic proteins such as the B-cell lymphoma 2 protein-associated agonist cell death protein (BAD).

Overall, obesity impacts the efficacy of multiple chemotherapeutic agents used in ALL therapy. Further research is needed to understand how the expanding adipose tissue in obesity disrupts chemotherapies, as these mechanisms can be used as potential therapeutic entry points to enhance treatment success.

### 4.3. Cytokines

Cytokines provide stimuli for the survival, differentiation, and proliferation of hematopoietic cells [95]. In obesity, the secretion of multiple cytokines including IL-3, IL-7, IL-8, TNFα and chemokines including monocyte chemotactic protein-1 (MCP-1, also called Chemokine C-C motif ligand 2 (CCL2)), are upregulated [67,96,97]. However, the potential role of obesity-driven cytokine upregulation in ALL pathogenesis is currently unclear.

While elevated IL-6, IL-17, and IL-18 levels at birth have been associated with increased risk of developing B-cell ALL later in life, their exact role in ALL development, if any, is unknown [67,96]. Elevated levels of TNFα and CCL2 in the cerebrospinal fluid of patients with ALL were initially linked to a higher risk of relapse, but these levels were later found to be related to intrathecal chemotherapy [98,99,100,101].

IL-7 is an essential cytokine in the maturation of T-cell ALL [102,103]. While IL-7 is a critical molecule in the development of both B- and T-cells in mice [104], human studies have revealed its impact on T-cell development only [105]. Notably, gain-of-function mutations of the IL-7 receptor gene (IL-7R) have been reported in children with T-cell ALL [102].

Both T-cells and IL-7 expression are most prominent in the thymus and bone marrow. IL-7 signaling through the IL-7 receptor activates the Janus kinase/signal transducer and activator of transcription protein 5 (JAK/STAT5) and the phosphatidylinositol 3-kinase/protein kinase B/mammalian target of rapamycin (PI3K/Akt/mTOR) pathways. The activation of the JAK/STAT5 pathway lowers naïve CD4 and CD8 T-cell numbers through unknown mechanisms [106]. PI3K catalyzes the production of phosphatidylinositol 3,4,5 triphosphate (PIP3), which allows for Akt and mTOR activation (Figure 1). In vitro studies with human T-cells demonstrate that this pathway is activated by IL-7 and is required for the proliferation and survival of both normal immature T-cells and ALL cells [103].

In the murine bone marrow, IL-3 and IL-7 promote B-cell proliferation and survival [107,108] and CXCL12 is essential for expanding leukemia cells [109]. In vitro studies revealed that IL-3 and IL-7 have clear synergistic responses to B-cell ALL proliferation when CXCL12 is present with either or both cytokines. These studies have also shown the upregulation of the PI3K and the p38 mitogen-activated protein kinase (p38MAPK) pathways and the promotion of B-cell ALL proliferation [107].

In summary, the role of cytokines in leukemia cell survival and proliferation makes IL-3 and IL-7 potential therapeutic targets for obesity-driven B-cell ALL, while IL-7 alone may be a target for obesity-driven T-cell ALL.

### 4.4. Adiponectin, Leptin, and Leptin Receptor

The most abundant adipokines that may influence cancer development and progression include adiponectin and leptin (Figure 1) [110,111].

Adiponectin circulates in an inverse proportion to adiposity and modulates several metabolic processes, including glucose homeostasis [96,112]. Hypoadiponectinemia is a phenomenon that was noted in ALL development and relapse [111,113]. Adiponectin activities are, in part, mediated by AMP-activated protein kinase (AMPK) that suppresses leukemia cell activity by stimulating tumor suppressors p53 and p21. The low levels of adiponectin in obesity may reduce the tumor suppressors’ activity, which may theoretically promote ALL development [113].

Adiponectin also exerts anti-inflammatory actions by downregulating the secretion of proinflammatory cytokines TNFα and IL-6 from activated macrophages, while stimulating the production of IL-10, an anti-inflammatory cytokine [114,115,116]. More research is needed to discern if this adipokine is associated explicitly with ALL development.

Leptin is a peptide hormone primarily produced by the adipose tissue [117]. Leptin impacts a wide range of biological activities, including the modulation of immune responses and angiogenesis [67,118]. In patients with obesity, while the leptin levels are increased, leptin receptor (LEPR) signaling is attenuated [119,120,121]. Leptin stimulates the production of several proinflammatory cytokines, including TNFα, IL-1β, IL-6, and IL-12, thus promoting chronic inflammation [67,78,122]. Furthermore, attenuated LEPR signaling has been linked to poor patient outcomes [120]. Specifically, B- and T-cell ALL have demonstrated faster progression and lower overall survival rates in LEPR-deficient mice models [120]. The mechanisms responsible for this association remain unclear.

Leptin also promotes angiogenesis in hematologic cancers. There is an increase in microvessel density within the bone marrow of patients with ALL, which suggests that hematological malignancies depend on angiogenesis for development and progression [123,124]. Competitive inhibition of leptin binding markedly decreases bone marrow microvessel density in rat models of leukemia [124]. The inhibition of leptin signaling in the bone marrow in ALL, and its impact on outcomes, needs further evaluation.

### 4.5. Insulin and Insulin-Like Growth Factor-1 (IGF-1)

Obesity drives the rise in insulin resistance and hyperinsulinemia [125], which is associated with hyperglycemia and dyslipidemia [96,126].

Insulin has mitogenic effects via its signaling pathway that is dysregulated in ALL (Figure 1) [127]. This signaling includes the overexpression of PI3K/Akt and STAT3 in lymphoblasts that is associated with chemotherapy resistance and poor prognosis [96,128,129,130,131,132]. For example, insulin reduces daunorubicin-induced toxicity and minimizes daunorubicin, vincristine, and L-asparaginase-mediated apoptosis of ALL cells in vitro [130].

Insulin-like growth factor-1 (IGF-1) is a growth factor that has a role in the progression of solid and hematologic malignancies [133]. In obese individuals, total IGF-1 levels are often in the normal to low range. However, free and active IGF-1 levels are generally higher than in nonobese individuals [96,134]. IGF-1 signaling activates many of the same pathways as insulin, including PI3K/Akt and MAPK. In vitro studies have shown that the stimulation of the IGF-1 receptor results in the proliferation of T-ALL cells and leads to chemotherapy resistance [135,136]. The role of IGF-1 in ALL pathogenesis and prognosis requires further evaluation.

### 4.6. Natural Killer Cell Dysregulation

Natural Killer (NK) cells are a lymphocyte subset that are vital for metabolic regulation and protection against infection and malignancy [137]. NK cells are implicated in innate immune responses and provide rapid reactions to environmental threats.

NK cell mechanisms linked directly to ALL are not reported in the literature to date. However, NK cells from obese humans are reduced in numbers [138] with lower cytotoxic activity than those from non-obese controls [139,140,141,142].

In high-fat-fed obese ALL mice, NK cells upregulate the secretion of cytokines and chemokines, including IL-2, IL-8, IL-12, and the chemokine macrophage inflammatory protein-1α (MIP-1α), which recruits macrophages to the adipose tissue (Figure 1) [137,143]. Human studies have also demonstrated that NK cells from obese individuals kill fewer tumor cells in vitro [139]. The role of NK cells in childhood ALL development requires further investigation.

### 4.7. Adipose-Derived Stem Cells

Adipose-derived stromal/stem cells (ASCs) are a type of mesenchymal stem cell found in the stromal vascular fraction of the adipose tissue [144,145,146]. ASCs secrete factors that are potentially associated with cancer development and progression, including IGF-1, transforming growth factor beta 1 (TGFβ1), vascular endothelial growth factor (VEGF), hepatocyte growth factor (HGF), and IL-8 [147,148,149,150,151].

Emerging evidence indicates that ASCs may promote the progression of hematologic cancers. Both B- and T-ALL cell lines expand in the presence of co-injected ASCs in mouse models, suggesting that ASCs promote the proliferation of ALL cells, and this proliferation occurs in a dose-dependent manner [78,152,153,154,155,156,157]. ASCs have also been investigated for their immunosuppressive properties, which may promote hematologic malignancies [153]. ASCs suppress immunity via interactions with innate immune cells including NK cells and dendritic cells as well as adaptive immune cells such as B- and T-lymphocytes. ASCs reduce NK cell proliferation and cytotoxic effects and inhibit the natural maturation of dendritic cells into B- and T-lymphocytes [78,158,159,160]. ASCs may be potentially important players in the initiation and propagation of ALL, and this area is in need of further research.

### 4.8. Glucose, Free Fatty Acids, and Amino Acids as Fuels for Leukemia Cells

The rapid proliferation of cancer cells demands large amounts of energy and nutrients. Adipocytes play an essential role in triglyceride storage and may fuel ALL cells with FFAs and amino acids (AAs).

The bone marrow provides the primary microenvironment for the development of leukemia. Increased expression of adipogenic genes CCAAT/enhancer-binding protein (CEBP) and peroxisome proliferator-activated receptor γ (PPARγ) have been observed in isolated mesenchymal stem cells from bone marrow aspirates of pediatric ALL patients, suggesting that the bone marrow is an active site of adipose tissue action [161].

In obesity, there is an excess of fuel storage within the expanded adipose tissue, both peripherally and in the bone marrow adipose depots. The higher glucose, FFA, and AA availability in both locations provides fuel substrates for nearby leukemia cells and may, in theory, support more rapid cell proliferation (Figure 1) [96,162].

ALL cells comply with the Warburg effect, with increased glucose uptake and glycolysis [163]. A study that targeted glucose transport found that B-ALL cells rely on GLUT1 for aerobic glycolysis and anaerobic metabolism [164]. Conditional GLUT1 gene knockout reduced cell proliferation, promoted apoptosis and increased sensitivity to Dasatinib in vivo [164]. Another possible mechanism reported in AML is the disruption of glucose metabolism by the induction of adipose tissue insulin-like growth factor binding Protein-1 production that impairs insulin sensitivity. Also, the depletion of gut serotonin and glucagon-like peptide-1 reduces insulin secretion, thus increasing the glucose available for leukemia cells to enhance their proliferation [165]. Although the same effect has yet to be demonstrated in ALL, this may be a plausible mechanism that requires further assessment.

Although complete leukemia remission rates were lower with hyperglycemia in adults, post-induction hyperglycemia in pediatric patients did not significantly influence outcomes [166]. Explanations for the difference in hyperglycemia effects between the groups include a higher incidence of comorbidities, increased systemic exposure to dexamethasone, and increased insulin resistance in the adult ALL population. Overall, the impact of higher glucose levels with obesity in pediatric ALL remains unclear.

Leukemia cells use FFAs as a source of energy for their proliferation and lipogenesis [96]. Overexpression of lipogenic enzymes by bone marrow adipocytes, such as hormone-sensitive lipase, and fatty acid transporters, such as fatty acid-binding protein-4 [167] and lipoprotein lipase [168], have been shown when the adipocytes were cocultured with acute myeloid and chronic lymphoblastic leukemia cells [167,168,169]. However, this overexpression and its effects have not yet been demonstrated with ALL cells.

Of note, when adipocytes were cocultured with human and murine ALL cells, the leukemia cells stimulated adipocytes to release FFAs, and ALL cells then incorporated the FFAs into phospholipid membranes and lipid droplets. [170]. A mechanism is proposed whereby adipocytes contribute to ALL development by increasing available FFAs for proliferation, though further studies would need to demonstrate this effect in vivo.

AAs are a vital source of fuel for cells that are also produced by adipocytes. ALL cells lack asparagine synthetase, which is necessary to synthesize essential AAs, including glutamate and aspartate from glutamine and asparagine, respectively. ALL cells depend on exogenous sources of AAs for cell cycle progression. Asparaginase treatments, such as L-asparaginase, hydrolyze and reduce the amount of asparagine, and to a lesser extent glutamine, available to ALL cells [171]. The bone marrow adipocytes may contribute to L-asparaginase resistance by producing high levels of glutamine in the interstitial fluid in the presence of ALL cells [172].

## 5. Targeted Therapy for Obesity-Associated Acute Lymphoblastic Leukemia

While obesity may be associated with the harboring of leukemia cells in the adipose tissue and circumventing chemotherapeutic efficacy, yet most of these effects are likely reversible if the adipose mass is decreased. For example, obese mice with ALL had an enhanced response to vincristine and improved survival when switched to a low-fat diet before chemotherapy initiation [173]. Therefore, targeting obesity in children at diagnosis, as well as during and post-therapy may improve treatment outcomes.

Several attempts to prevent and manage obesity during and after ALL therapy have been undertaken. However, there is a lack of high-evidence, and no effective lifestyle interventions to treat and prevent obesity in children with ALL are currently available [174]. There is a need for rigorous evidence to manage obesity in survivors and assess its impact on outcomes.

One crucial pathway in ALL is the mTOR pathway, which plays a critical role in the development and progression of both B- and T-cell ALL [128]. Significantly, mTOR is activated through several mechanisms in obesity. Multiple factors, including insulin, IGF-1, IL-1β, IL-6, IL-17 and TNFα can activate the PI3K/Akt/mTOR pathway and promote oncogenesis. There are several mechanisms through which this pathway impacts ALL development, including increased genomic instability via blocking of checkpoint kinase 1, increased mammalian target of rapamycin complex 1 (mTORC1) activity, enhancing protein synthesis, activating hypoxia-inducible factor-1α and increasing VEGF for angiogenesis promotion [97].

Several available compounds target mTOR and can provide potential therapeutic benefits. One such compound is metformin [96]. This medication, traditionally used to improve insulin sensitivity in patients with type 2 diabetes, stimulates AMPK which is a potent inhibitor of mTOR [175]. In T-cell ALL specifically, metformin induces autophagy and apoptosis of leukemia cells [176]. In vitro studies have shown that metformin is effective in downregulating signaling, with dephosphorylation and marked inhibition of mRNA translation in T-ALL cells [114]. Metformin triggers cell growth arrest and apoptosis by inducing endoplasmic reticulum stress with unfolded protein response-mediated cell death pathways. The mTOR pathway is impacted explicitly by an increase in unfolded and misfolded proteins within the endoplasmic reticulum, which results in increased proteotoxicity and cell death. While the utility of metformin in relapsed pediatric ALL patients has undergone phase 1 studies for review of toxicity, with tolerability at an appropriate dose of 1000 mg/m^2^/day [177], the effects of metformin on responses to therapy and outcomes are not yet established. This is an important area of further study.

## 6. Conclusions

Childhood obesity is an ongoing epidemic, and there is extensive evidence to support its health risks throughout the lifespan. Our current understanding of the potential mechanisms through which obesity may initiate and promote ALL development and propagation is limited. However, there are conceivable pathways whereby a general increase in body mass, or specific expansion of the adipose tissue, may contribute to ALL genesis. There is more definite evidence for the long-term implications of ALL treatment on pediatric obesity in survivors.

Future research efforts should use in-vitro and in-vivo approaches to understand the impact of the upregulation of inflammatory and mitogenic pathways, such as mTOR, in obesity on leukemia genesis and propagation. Furthermore, the role of the adipose microenvironment and adipocytokines in proliferation and protection of leukemia cells needs further analysis to see if the adipose tissue provision of a sanctuary for leukemia cells can be prevented. The role of other tissue compartments, such as skeletal muscle and muscle-based intermuscular adipose tissue in protecting leukemia cells from chemotherapies, and providing a protective microenvironment for lymphoblasts to propagate, is also unknown, and this area of research has not yet been explored. The interrogation of potential cytokines as therapeutic targets, including IL-3 and IL-7, is also an area of further potential investigation.

Although multiple potential mechanisms may be implicated in driving ALL in obesity, more evidence is urgently needed to establish if obesity can directly cause ALL. For now, there is no clear proof that obesity is causal in pediatric ALL.

## Figures and Tables

**Figure 1 cancers-12-03082-f001:**
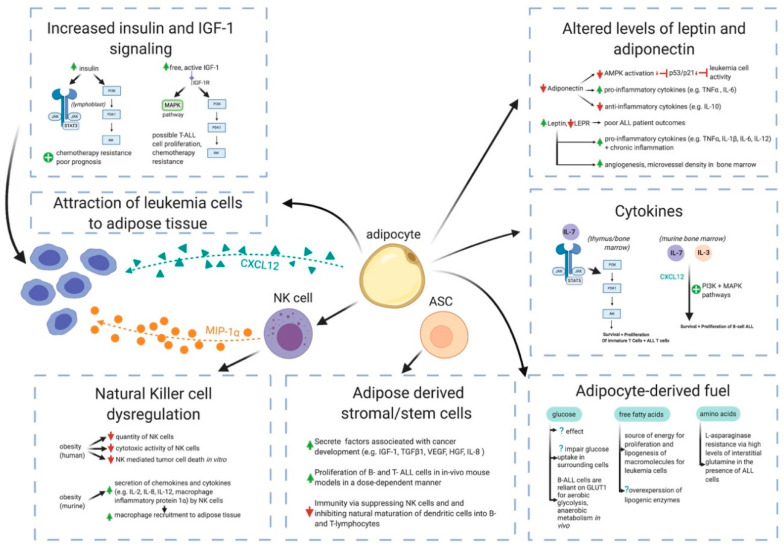
Potential mechanisms for obesity associations with ALL. The adipocytes secrete chemokines that can attract leukemia cells, including cysteine-X-cysteine (C-X-C) motif chemokine 12 (CXCL12). Adipose tissue-infiltrating Natural Killer (NK) cells secrete macrophage inflammatory protein-1 alpha (MIP-1α), another chemokine for leukemia cells. Obesity is also associated with reduced numbers and cytotoxic activity of NK cells. Adipose-derived stromal/stem cells (ASC) secrete factors that promote the proliferation of lymphoblasts.

**Table 1 cancers-12-03082-t001:** Rates of Obesity at Acute Lymphoblastic Leukemia (ALL) Diagnosis, End of Treatment, and During Follow-up.

Author (Reference)	Study Design	Populations	Protocols	Findings
Van Dongen-Melman et al. [34]	RCR	*n* = 113 Age: 0.5–15 years		Overweight/obese patients (>90th BMI percentile) constituted 7.9% (*n* = 9) of the sample at diagnosis. At the end of therapy, 30% (*n* = 34) were overweight/obese. At four years after treatment completion, 23.9% (*n* = 27) were overweight/obese. Radiotherapy was not associated with obesity. Patients who received a combination of dexamethasone and prednisone were at the highest risk of being obese (44%). Higher cumulative steroid dose did not contribute to more obesity.
Withycombe et al. [35]	RCR	*n* = 1638 Age: 2–20 years	COG (CCG 1961)	Obesity rates in children with high risk ALL were 14% at baseline and 23% at the end of treatment. Females, Black or Hispanic, and age 5–9-year-old, but not cranial irradiation, were risk factors. The highest increase in BMI% was between maintenance phases 1–3, 9–12 months postdiagnosis. (Induction BMI% 7.2, BMI% 12.6 by maintenance phase 3).
Breene et al. [36]	RCR	*n* = 77 Age: 1–16 years	MRC UKALL97 protocol	Whole group: Patients received only chemotherapy and no CRT. Thirty patients (39%) received prednisolone, and 47 (61%) got dexamethasone. Weight gain was not linked to steroids. There was a significant rise in BMI-SDS from diagnosis (0.35, 95% CI 0.20–0.50) to the end of treatment (1.29, 95% CI 1.13–1.45, *p* < 0.0001), and at three-year follow-up (1.04, 95% CI 0.85–1.22, *p* <0.0001). More survivors were overweight or obese at three years post treatment (25/53, 47.20%) when compared to diagnosis (23/77, 29.90%), (*p*-value 0.01) Female subgroup: Significant rise in BMI-SDS from diagnosis (0.46, 95% CI 0.27–0.64), at end of treatment (1.46, 95% CI 1.26–1.66, *p* < 0.0001) and at three-year follow-up (1.24, 95% CI 1.03–1.45, *p* < 0.0001) Male subgroup: Significant rise in BMI-SDS from diagnosis (0.24, 95% CI 0.01–0.46), to end of treatment (1.11, 95% CI 0.85–1.36, *p* < 0.0001) and at three-year follow-up (0.77, 95% CI 0.43–1.10, *p* < 0.0001)
Razzouk et al. [37]	RCS	*n* = 248 Age < 19 years At diagnosis, 13.2–30 years at the latest assessment	Chemotherapy-Total Therapy Study X protocol	The prevalence of overweight/obesity in 0–6-year-old (6%) was lower than that in the 13–19-year-old group (19%) at diagnosis. At adult height attainment, the prevalence of overweight/obese in 0-6 years of age at diagnosis was 41%, versus 13–19 years of age at 35%. These rates are close to the USA’s general population. Those <6 years of age (OR 2.3, 95% CI 1.2–4.2, *p*-value 0.01), male (OR 0.50, 95% CI 0.28–0.91, *p*-value 0.02), and being overweight/obese at diagnosis (OR 14.00, 95% CI 5.00–39.00, *p*-value < 0.0001) were predictors of obesity at adult height attainment. CRT (24 Gy) led to an increase in BMI trajectory, but this was not different from the 18 Gy CRT group.
Ghosh et al. [38] ^*^	RCS	*n* = 4775 Age: 2–30 years Age, sex, and ethnicity-matched controls from NHANES	COG AALL17D2	Newly diagnosed had overweight rates of 17% and obesity rates of 20%. 58% had an average weight, and 5% were underweight. Males, Hispanics, and B-cell ALL were associated with obesity. Obesity was associated with CNS disease.
Foster et al. [39]	RCS	*n* = 121 Age: 2–15 years	COG AALL0232, AALL0331, AALL0932, AALL1131, POG 9904, POG9905	15% of patients were overweight and 15% of patients were obese at the time of diagnosis of ALL. At 5-year follow-up, 22% of patients were overweight and 35% of patients were obese. Start of treatment BMI z-score 0.25 (95% CI 0.01–0.49) Five-year follow-up BMI z-score 0.99 (95% CI 0.79–1.19; *p*-value < 0.0001)
Didi et al. [40]	PCS	*n* = 114 Age: 2–16 years	MRC UKALL protocol	23/51 male (45%) and 30/63 female (47%) patients were obese at final height attainment. Female patients: Girls’ obesity occurred from start to end of treatment then plateaued. Start of treatment BMI z-score 0.05 (95% CI −2.2, 2.0) End of treatment BMI z-score 1.2, 95% CI 0.3–2.8; *p*-value 0.0002) Male patients: Boys gained weight from the start of treatment, and weight gain continued post-treatment completion. Start of treatment BMI z -score 0.10 (95% CI −1.1, 1.3); End of treatment BMI z-score 0.6 (95% CI −0.4, 4.9; *p*-value 0.001)
Craig et al. [41] ^#^	CSS	*n* = 213 radiotherapy, *n* = 85 no radiotherapy Age: 0–16 years	Unirradiated: MRC UKALL XI or infant ALL protocol Irradiated: 1 Great Ormond Street Hospital devised protocol, MRC UKALL III-VI or UKALL VIII-X if receiving 18–20 Gy CRT 2 Great Ormond Street Hospital devised protocol, UKALL I–VI, and UKALL VIII–X if received 22–24 Gy CRT	Female 18–20 Gy CRT: BMI z-score at diagnosis −0.24 ± 0.15 BMI z-score at end of treatment 0.46 ± 0.11 (*p*-value < 0.0001) 22–24 Gy CRT: BMI z-score at diagnosis −0.70 ± 0.16 BMI z-score at the end of treatment −0.17 ± 0.12 (*p*-value 0.0005) Male 18–20 Gy CRT: BMI z-score at diagnosis −0.40 ± 0.16 BMI z-score at end of treatment 0.37 ± 0.16 (*p*-value < 0.0001) 22–24 Gy CRT: BMI z-score at diagnosis −0.17 ± 0.28 BMI z-score at end of treatment 0.48 ± 0.16 (*p*-value 0.02) BMI z-scores in patients with no history of CRT: Female BMI z-score at diagnosis −0.12 ± 0.19 BMI z-score at end of treatment 0.70 ± 0.17 (*p*-value < 0.0001) Male BMI z-score at diagnosis 0.23 ± 0.26 BMI z-score at end of treatment 0.81 ± 0.18 (*p*-value 0.01)

Abbreviations: RCR, retrospective chart review; BMI, body mass index; COG, Children’s Oncology Group; CCG, Children’s Cancer Group; BMI%, body mass index percentile; MRC, Medical Research Council; UKALL, United Kingdom Acute Lymphoblastic Leukemia Regimen; NR, not reported; CI, confidence interval; SDS, standard deviation score; NS, not significant; RCS, retrospective cohort study; CRT, cranial irradiation; OR, odds ratio; Gy, Gray; COG AALL, Children’s Oncology Group Acute Lymphoblastic Leukemia protocols; POG, Pediatrics Oncology Group; z-score, standard score; PCS prospective cohort study; CCS, cross-sectional study. ^#^ The *p*-values include a comparison of BMI z-scores from baseline to end of treatment. ^*^ This abstract reported a study that compared ALL patients to the National Health and Nutrition Examination Survey (NHANES) control subjects (*n* = 30,107). This abstract also validated the already known association of being underweight at ALL diagnosis.

**Table 2 cancers-12-03082-t002:** Birth Weight and the Risk of Development of ALL.

Author (Reference)	Study Design	Populations	Findings
Hjalgrim et al. [43]	Meta-analysis	18 studies *n* = 10,282 Age: 0–29 years	BWt > 4000 was associated with a trend of higher risk of ALL (OR 1.26, 95% CI 1.17–1.37)
Caughey et al. [44]	Meta-analysis	32 studies *n* = 16,501 Leukemias (*n* = 10,974 ALL) Age: ≤ 30 years	Significant odds for the association of high BWt with ALL risk (OR 1.23, 95% CI 1.15–1.32)
Milne et al. [45]	CCS	*n* = 519 patients Age: 0–14 years	OR for 1 SD increase in proportion to optimal BWt 1.18 (95% CI 1.04–1.35, p < 0.05). However, faster fetal growth, instead of BWt, was the factor associated with ALL risk
Jiménez-Hernández et al. [46]	CCS	*n* = 2910 children Age: 0–18 years	ALL is associated with a birth weight ≥ 2500 g (OR 2.06, 95% CI 1.59–2.66) Birth weight ≥ 3500 g was also associated with ALL OR 1.19 (95% CI: 1.00–1.41)
Sprehe et al. [47]	RCR	*n* = 13,988 children Age < 5 years at cancer diagnosis + age-matched controls	Increased risk of ALL was associated with LGA compared to AGA LGA (<4000g) OR 1.5 (95% CI: 0.97–2.52, *p* = 0.0005) LGA (>4000g) OR 1.67 (95% CI: 1.29–2.16, *p* = 0.0005)

Abbreviations: BWt, birth weight; OR, odds ratio; CI, confidence interval; SD, standard deviation; CCS, case-control study; LGA, large for gestational age; AGA, appropriate for gestational age; RCR, retrospective chart review.

**Table 3 cancers-12-03082-t003:** Meta-Analyses of Relapse Risk with Obesity in ALL Patients.

Author	Study Design	Population	Protocol	Findings
Saenz et al. [57]	RCS + Meta-analysis	*n* = 181 Age: 2–17 years	COG	A trend of relapse risk for obese/overweight patients (HR 2.89, 95% CI 0.89–9.36, p = 0.08) in age-and sex-adjusted patients ≥ 10 years old. While meta-analyses reported increased mortality in the overweight/obese group (HR 1.39, 95% CI 1.16–1.46, p < 0.05), this was not confirmed in adjusted analyses
Orgel et al. [58]	Meta-analysis	11 articles *n* = 8680 (ALL), Age: 0–21 years	Included all treatment regimens that reported the effect of weight on treatment outcomes, specifically EFS mortality, overall survival, the cumulative incidence of relapse, and treatment-related toxicity	When compared to a lower BMI, higher BMI was associated with a statistically nonsignificant trend of risk of relapse (RR 1.17; 95% CI: 0.99, 1.38) Patients with a higher BMI had lower EFS (RR 1.35, 95% CI 1.20, 1.51) than those with lower BMI

Abbreviations: RCS, retrospective cohort study; COG, Children’s Oncology Group; HR, hazard ratio; CI, confidence interval; ALL, Acute Lymphoblastic Leukemia; EFS, event-free survival; RR, risk ratio; BMI, body mass index.

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
