# Peer review of "Is There a Causal Relationship between Childhood Obesity and Acute Lymphoblastic Leukemia? A Review"

_cancers, 2020, doi:10.3390/cancers12113082_

Round 1

Reviewer 1 Report

The manuscript is an interesting and comprehensive review on the causal relationship between childhood obesity and Acute Lymphoblastic Leukemia (ALL). The article is well written with a good structure that makes it very easy to follow and understand. The authors discuss, very thoroughly, several potential molecular mechanisms by which obesity can drive and support ALL pathogenesis. The overall message of the manuscript is clear and concise, relationships between mechanisms, concepts and cellular entities are easily understood. Importantly, the authors use as references the most prominent and recent studies of the field, as 53 out of 161 references are studies that have 5 years or less. Moreover, 21 out of 161 have 2 years or less suggesting an up-to-date integration with the current knowledge in the field.

Nevertheless its significance and value for the field some minor issues should be addressed:

Minor comments:

  • The paragraph between lines 63 and 66 is very confusing, please rephrase it.
  • The paragraph between lines 103 and 108 is in italic, any reason for this?
  • In line 309 asessment is misspelled, it should be assessment.
  • A short query on google retrived three articles that are not cited in the manuscript and the authors should include these references and incorporate them in the text: i) Yun et al, 2010 (DOI: 10.1158/1940-6207.CAPR-10-0087); ii) Meenan et al, 2019 (DOI: 1002/pbc.27515); and iii) Foster et al, 2019 (DOI: 10.1371/journal.pone.0217932).
  • The text for heading 5 is very concise regarding the pivotal role of mTOR on metabolism and ALL.
  • The conclusion part of the review is also very concise and does not provide any speculation on studies that should be perform to address the role of obesity as a potential cause of ALL.

Author Response

Reviewer 1:

Comments and Suggestions for Authors

The manuscript is an interesting and comprehensive review on the causal relationship between childhood obesity and Acute Lymphoblastic Leukemia (ALL). The article is well written with a good structure that makes it very easy to follow and understand. The authors discuss, very thoroughly, several potential molecular mechanisms by which obesity can drive and support ALL pathogenesis. The overall message of the manuscript is clear and concise, relationships between mechanisms, concepts and cellular entities are easily understood. Importantly, the authors use as references the most prominent and recent studies of the field, as 53 out of 161 references are studies that have 5 years or less. Moreover, 21 out of 161 have 2 years or less suggesting an up-to-date integration with the current knowledge in the field.

Response: We would like to thank the reviewer for their positive comments and for their thorough review.

Nevertheless its significance and value for the field some minor issues should be addressed:

Minor comments:

  • The paragraph between lines 63 and 66 is very confusing, please rephrase it.

Response: This section was re-written (lines 79-89).

  • The paragraph between lines 103 and 108 is in italic, any reason for this?

Response: This text formatting issue has been rectified. There was no specific reason for the italicization of text.

  • In line 309 asessment is misspelled, it should be assessment.

Response: This misspelling has been corrected.

  • A short query on google retrieved three articles that are not cited in the manuscript and the authors should include these references and incorporate them in the text: i) Yun et al, 2010 (DOI: 10.1158/1940-6207.CAPR-10-0087); ii) Meenan et al, 2019 (DOI: 1002/pbc.27515); and iii) Foster et al, 2019 (DOI: 10.1371/journal.pone.0217932).

Response: We thank the reviewer for their comment. The articles proposed are all excellent articles and we have added them to our review.

Foster et al is a relevant and recent retrospective cohort study that describes trends in obesity at diagnosis and following pediatric patients who are treated with strictly chemotherapy for ALL. This reference has been added (Line 78, reference #41) and to Table 1.

Meenan et al is a large retrospective chart review the describes the increased risks of morbidity in children undergoing treatment for ALL who are obese. This reference has been added  (Lines 132-136, reference #60).

Yun et al is a pre-clinical trial examining the effects of obesity and high-fat diets on the progression of ALL in mouse models. They demonstrated a more rapid progression of ALL in the high-fat-fed obese mice. This reference has been added (Line 157, reference #72). Thank you for these supportive references.

  • The text for heading 5 is very concise regarding the pivotal role of mTOR on metabolism and ALL.

Response: We thank the reviewer for their comment. We have re-written and added references to this section (Lines 383-403)

  • The conclusion part of the review is also very concise and does not provide any speculation on studies that should be performed to address the role of obesity as a potential cause of ALL.

Response: We thank the reviewer for their comment. We have expanded the future directions section under conclusions (lines 405-422).

Reviewer 2 Report

The manuscript by Dushnicky et al entitled "Is There a Causal Relationship between Childhood Obesity and Acute Lymphoblastic Leukemia? A Review” explore the role and significance of Childhood Obesity and Acute Lymphoblastic Leukemia. Specifically, they try to link relationship between endocrine, metabolic, and immune dysregulation triggered by obesity and its potential role in the genesis of pediatric ALL. Furthermore, they describe possible mechanisms such as adipose tissue attraction and protection of lymphoblasts and its impact on pharmacokinetics of ALL chemotherapies. In this article authors collected vast information from review literature and very nicely combined in the manuscript.

Overall this is an interesting review article with great potential. Collectively, this study conducted meticulously and appears to be well designed and the collected information support the conclusion. The review literature are clearly presented, and appears solid and their analyses are reasonable. Simultaneously, I have some minor suggestions need to be addressed by authors.

Comments:

  1. In the introduction section, background information is not adequate. It is not easy to follow the motivation of this study. To filling this gap of information, need some literature about how metabolic, obesity plays an important role in progression of cancer in preclinical models. (See, PMID: 29568521 and PMID: 27980732).
  2. The authors need to rethink about how to present their view and opinions on this topics. A review is to present your scientific conclusion or hypothesis after analyzing current available data, not just pile up existing data and trials. Wish the authors first ask ourselves what you intend to tell the audience then organize a more concentrated, short, but hitting the point review.
  3. Are some trials that investigate the effect of obesity modulators in clinical practice in ALL? The authors should discuss potential treatment strategies of obesity modulators.
  4. Authors incorporated a lot of information in the introduction section from the literature. Please incorporated only those information which is related to this review. Please remove unnecessary literature from this review.
  5. It would be very nice, if authors could write the attractive titles of each starting paragraphs.
  1. The manuscript is poorly written, and requires an attention to improve punctuations, grammar and the readability. Spacing of the words need to be thoroughly checked throughout the manuscript. There are many places, authors should correct the typo errors.
  1. There are many places authors could slightly modify the sentences and no need to write very broad sentences.
  2. These kinds of studies have limitations. Hence, the author should have highlighted the potential limitations and shortcomings and suggests what could be the next step in this area of research.

Author Response

Reviewer 2:

Comments and Suggestions for Authors

The manuscript by Dushnicky et al entitled "Is There a Causal Relationship between Childhood Obesity and Acute Lymphoblastic Leukemia? A Review” explore the role and significance of Childhood Obesity and Acute Lymphoblastic Leukemia. Specifically, they try to link relationship between endocrine, metabolic, and immune dysregulation triggered by obesity and its potential role in the genesis of pediatric ALL. Furthermore, they describe possible mechanisms such as adipose tissue attraction and protection of lymphoblasts and its impact on pharmacokinetics of ALL chemotherapies. In this article authors collected vast information from review literature and very nicely combined in the manuscript.

Overall this is an interesting review article with great potential. Collectively, this study conducted meticulously and appears to be well designed and the collected information support the conclusion. The review literature are clearly presented, and appears solid and their analyses are reasonable. Simultaneously, I have some minor suggestions need to be addressed by authors.

Response: We thank the reviewer for their supportive comments.

Comments:

  1. In the introduction section, background information is not adequate. It is not easy to follow the motivation of this study. To fill this gap of information, need some literature about how metabolic, obesity plays an important role in progression of cancer in preclinical models. (See, PMID: 29568521 and PMID: 27980732).

Response: We thank the reviewer for their insightful comment. We have added the papers highlighted by the reviewer to the manuscript (Lines 63-68, references 30,31).

Malvi et al, describe the impact of obesity on dacarbazine therapy in melanoma, namely the role of leptin and resistin in promoting the growth of melanoma cells and reducing the efficacy of dacarbazine. In another paper, Malvi et al describe how diet-induced obesity not only promotes melanoma development but also reduces the ability of dacarbazine to access melanoma cells.

We believe that drawing parallels between ALL and melanoma is quite helpful to the reader to demonstrate our points. While the literature on the impact of obesity on cancer is vast, we believe that the approach proposed by the reviewer is most appropriate. Thank you.

  1. The authors need to rethink how to present their view and opinions on this topics. A review is to present your scientific conclusion or hypothesis after analyzing current available data, not just pile up existing data and trials. Wish the authors first ask ourselves what you intend to tell the audience then organize a more concentrated, short, but hitting the point review.

Response: We thank the reviewer for their comments. In the revised manuscript, we have read through and added the conclusions for different sections in the paper.

Some examples to address the reviewer’s point are noted below, with few others in the manuscript:

  • We summarize the association of age, sex, and leukemia type with obesity at diagnosis (Lines 84-89)
  • Birth weight and ALL risk (Lines 109-111)
  • We conclude that there are other factors beyond steroids and radiation that drive obesity risk in ALL survivors (Lines 129-130).
  • We conclude that there is no evidence that obesity is associated with an increase in relapse risk in ALL (Lines 141).
  • We summarize the effect of obesity on chemotherapeutic pharmacokinetics and outlines future directions (Lines 204-207).
  1. Are some trials that investigate the effect of obesity modulators in clinical practice in ALL? The authors should discuss potential treatment strategies of obesity modulators.

Response: Our group has recently published a systematic review and meta-analysis that demonstrated the lack of high-quality evidence for effective lifestyle interventions in children with ALL. The interventions did not result in a change in BMI z-scores. We include this review and its conclusions in the manuscript (Lines 379-381, reference #174).

In addition, Phase 1 trials of metformin in pediatric patients with relapsed ALL is the only pharmacotherapy trial that has been documented to date. This is outlined in section 5 (Lines 391-403).

  1. Authors incorporated a lot of information in the introduction section from the literature. Please incorporated only those information which is related to this review. Please remove unnecessary literature from this review.

Response: We have shortened and focused the introduction on the primary question of this review.

  1. It would be very nice, if authors could write the attractive titles of each starting paragraphs.

Response: We thank the reviewer for their comment. We tried our very best to adjust titles to make them more informative and attractive. Two notable examples are reported below:

4.1. The White Adipose Tissue is Far More Than a Passive Calorie Sink in Obesity (Line 161)

4.2. Adipocytes Interference in Pharmacokinetics of ALL Chemotherapies (Line 178)

  1. The manuscript is poorly written, and requires an attention to improve punctuations, grammar and the readability. Spacing of the words need to be thoroughly checked throughout the manuscript. There are many places, authors should correct the typo errors.

Response: We apologize to the reviewer for these issues. We have gone back, and read the manuscript thoroughly to make the corrections. We have left track changes for the reviewer’s assessment. please do let us know if there are additional changes that are required.

  1. There are many places authors could slightly modify the sentences and no need to write very broad sentences.

Response: We have gone through the manuscript, and reviewed sentence structures, and modified them accordingly. Please let us know if there are any specific areas that remain in need of adjustment. We would be happy to address any further comments.

  1. These kinds of studies have limitations. Hence, the author should have highlighted the potential limitations and shortcomings and suggests what could be the next step in this area of research.

Response: We thank the reviewer for their comments, and we have added a section specifically to highlight future directions (Lines 411-422).

Reviewer 3 Report

In general this manuscript is interesting since there is a need for summarizing current evidence on the relationship between obesity and ALL. However I have the following comments to be addressed:

In the introduction section there are several sentences which require adequate references to previous works.

In my opinion, some tables And/or figures could improve the comprehension of this work. I would suggest to summarize current evidence using a table and visualize the major pathways involved using a figure.

Finally, I would suggest an English editing for some typos and confusing paragraohs

Author Response

Reviewer 3:

Comments and Suggestions for Authors

In general this manuscript is interesting since there is a need for summarizing current evidence on the relationship between obesity and ALL. However I have the following comments to be addressed:

In the introduction section there are several sentences which require adequate references to previous works.

Response: We thank the reviewer for their comment. We have added additional references to this section.

In my opinion, some tables And/or figures could improve the comprehension of this work. I would suggest to summarize current evidence using a table and visualize the major pathways involved using a figure.

Response: We thank the reviewer for their comment. We have modified the three tables so that studies are grouped by design and added more references. We also optimized the figure to encompass the paper conclusions. We believe that the information provided in this paper are providing significant additional material to inform readers of the content.

Finally, I would suggest an English editing for some typos and confusing paragraohs

Response: We thank the reviewer for this comment. We have gone back and read through the manuscript thoroughly and made significant changes to its contents. If there are specific areas for improvement, we would be very happy to address them further.

Round 2

Reviewer 3 Report

Tua authors addresses all my comments

This manuscript is a resubmission of an earlier submission. The following is a list of the peer review reports and author responses from that submission.

Round 1

Reviewer 1 Report

In general this manuscript is interesting since there is a need for summarizing current evidence on the relationship between obesity and ALL. However I have the following comments to be addressed:

In the introduction section there are several sentences which require adequate references to previous works.

In my opinion, some tables And/or figures could improve the comprehension of this work. I would suggest to summarize current evidence using a table and visualize the major pathways involved using a figure.

Finally, I would suggest an English editing for some typos and confusing paragraohs